# FlashI2V: Fourier-Guided Latent Shifting Prevents Conditional Image Leakage in Image-to-Video Generation

## Abstract

In Image-to-Video (I2V) generation, a video is created using an input image as the first-frame condition. Existing I2V methods concatenate the full information of the conditional image with noisy latents to achieve high fidelity. However, the denoisers in these methods tend to shortcut the conditional image, which is known as conditional image leakage, leading to performance degradation issues such as slow motion and color inconsistency. In this work, we further clarify that conditional image leakage leads to **overfitting** to in-domain data and decreases the performance in out-of-domain scenarios. Moreover, we introduce **F**ourier-Guided **La**tent **Sh**ifting **I2V**, named **FlashI2V**, to prevent conditional image leakage. Concretely, FlashI2V consists of: (1) **Latent Shifting.** We modify the source and target distributions of flow matching by subtracting the conditional image information from the noisy latents, thereby incorporating the condition implicitly. (2) **Fourier Guidance.** We use high-frequency magnitude features obtained by the Fourier Transform to accelerate convergence and enable the adjustment of detail levels in the generated video. Experimental results show that our method effectively overcomes conditional image leakage and achieves the best generalization and performance on out-of-domain data among various I2V paradigms. With only 1.3B parameters, FlashI2V achieves a dynamic degree score of 53.01 on Vbench-I2V, surpassing CogVideoX1.5-5B-I2V and Wan2.1-I2V-14B-480P.

## 1 Introduction

Conditional Video Generation (Ren et al., 2024; Zhang et al., 2023; Hu, 2024; Li et al., 2025a; Yuan et al., 2025b; Yu et al., 2024; Zhang et al., 2025) refers to the technology that generates videos based on user-provided conditions, with significant applications being Text-to-Video (T2V) Generation and Image-to-Video (I2V) Generation. Since T2V generation produces a video solely based on a prompt, it struggles to accurately define scenes, such as accurate color and shape within the video. In contrast, I2V generation creates a video from both a user-provided image and a descriptive prompt, ensuring that the video content semantically aligns with the prompt and the first frame matches the provided image at the pixel level. In the commercial State-of-the-Art (SOTA) video generation product, Kling (AI, 2025), 85% of usage calls are for I2V generation.

Leveraged in I2V methods including Stable Video Diffusion (SVD) (Blattmann et al., 2023), Open-Sora Plan (Lin et al., 2024a), CogVideoX (Yang et al., 2024), and Wan2.1 (Wan et al., 2025), existing approaches concatenate the conditional image latents encoded by a Variational Autoencoder (VAE) (Kingma & Welling, 2013) with the noisy latents along the channel dimension, achieving exceptionally high fidelity for the first frame. However, previous works (Zhao et al., 2024; Choi et al., 2025) highlight that existing methods suffer from conditional image leakage. Especially at large time steps, the denoiser directly utilizes the condition in a shortcut manner to minimize loss instead of performing the complex denoising process during training, resulting in slow motion in the generated output during inference. In addition to slow motion, we also observe other performance degradation issues such as color inconsistency in the generated video, as shown in Fig. 1a.

To investigate why conditional image leakage leads to performance degradation, we explore the generalization of the existing concatenating I2V paradigm. During training, the conditional image is

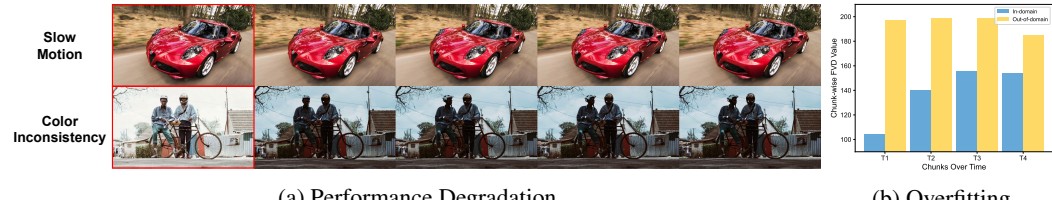

(a) Performance Degradation                  (b) Overfitting

Figure 1: **Conditional image leakage**. (a) Conditional image leakage causes performance degradation issues, where the videos are sampled from Wan2.1-I2V-14B-480P with Vbench-I2V text-image pairs. (b) In the existing I2V paradigm, we observe that chunk-wise FVD on in-domain data increases over time, while chunk-wise FVD on out-of-domain data remains consistently high, indicating that the law learned on in-domain data by the existing paradigm fails to generalize to out-of-domain data.

the first frame of a video. In contrast, during inference, the conditional image can come from any source and is not necessarily the first frame of an existing video. The ability to generate reasonable and high-quality videos from any conditional image requires strong generalization in I2V methods. Since we cannot achieve the training dataset of any existing model, we train a model with weights initialized from Wan-T2V-1.3B using the existing concatenating I2V paradigm and compare its performance on both in-domain and out-of-domain data. Each video is divided into temporal chunks with an equal frame interval. We then compare the Fréchet Video Distance (FVD) (Unterthiner et al., 2018) of the generated chunks with the ground truth chunks to assess the generation quality at different time points in the video. Theoretically, the first frame of the generated video must exactly match the conditional image, while subsequent frames lack such constraints, resulting in an increasing chunk-wise FVD over time. In a desired I2V paradigm, this increasing pattern should hold for both in-domain and out-of-domain data. As illustrated in Fig. 1b, experimental results reveal that chunk-wise FVD on in-domain data increases gradually over time. However, in out-of-domain data, chunk-wise FVD remains consistently high. By comparing the chunk-wise FVD variation patterns on in-domain and out-of-domain data, we conclude that even if the first frame matches the conditional image, shortcutting causes out-of-domain results to lack coherent video quality. The law learned from in-domain data fails to generalize to out-of-domain data, indicating that the concatenating paradigm faces an overfitting challenge, and a more reasonable paradigm is expected.

To prevent conditional image leakage, we propose a method that introduces conditions through **F**ourier-Guided **La**tent **Sh**ifting **I2V**, termed **FlashI2V**. The method consists of two parts: (1) **Latent Shifting**. Since flow matching imposes no restrictions on the source and target distributions, we encode the conditional image latents using a time-independent network and subtract the encoding from both the source and target distributions. The new velocity field is structurally the same as the velocity field in the original T2V model. The time-independent network is initialized to zero, ensuring that the input of the denoiser remains unchanged at the beginning of training. As a result, the denoiser gradually learns to utilize information from the conditional image through the shifted latents. Latent shifting requires recovering content from a mix of noisy latents and condition information. At larger time steps, the lower signal-to-noise ratio makes content recovery more difficult, which fundamentally prevents the leakage caused by shortcutting. (2) **Fourier Guidance**. Since the conditional image information needs to be recovered from the shifted latents, latent shifting requires more time and data to achieve first-frame fidelity comparable to existing methods. To accelerate convergence, we apply the Fourier Transform to extract high-frequency magnitude features from the conditional image latents and concatenate them with noisy latents. Since these magnitude high-frequency features only represent the relative strength of the signal, they serve as a supplement to latent shifting, which cannot lead to shortcutting. Moreover, by adjusting the cutoff frequency of the Fourier Transform, we can easily control the detail level in the generated video.

Compared to various existing I2V paradigms, only FlashI2V demonstrates the same FVD variation pattern on both in-domain and out-of-domain data, indicating that it avoids the leakage caused by shortcutting the conditional image. Furthermore, FlashI2V achieves the lowest FVD value among different I2V paradigms, showcasing its excellent performance. With only 1.3B parameters, FlashI2V achieves comparable scores to CogVideoX1.5-5B-I2V and Wan2.1-I2V-14B-480P on Vbench-I2V (Huang et al., 2024; Zheng et al., 2025) and obtains a dynamic degree score of 53.01, significantly outperforming the other two methods with larger parameter sizes.

In summary, our contributions are as follows: (1) By analyzing the chunk-wise FVD variation patterns in various existing I2V paradigms, we show that conditional image leakage causes overfitting to in-domain data, leading to performance degradation issues like slow motion and color inconsistency during inference. (2) We propose latent shifting, which implicitly introduces conditions based on flow matching characteristics. Additionally, we use high-frequency magnitude features from the Fourier Transform as guidance to accelerate convergence and enable the flexible control of detail levels in the generated video. (3) Experimental results show that FlashI2V exhibits the best generalization and performance across various I2V paradigms and effectively avoids overfitting caused by conditional image leakage. Specifically, with only 1.3B parameters, FlashI2V achieves a dynamic degree score of 53.01, surpassing CogVideoX1.5-5B-I2V and Wan2.1-I2V-14B-480P.

## 2 RELATED WORK

### 2.1 TEXT-TO-VIDEO GENERATION

In recent years, Text-to-Video Generation has made significant progress. T2V methods often use diffusion models (Ho et al., 2020; Song et al., 2020a;b) to model the generation process. Previous works typically employ UNet (Ronneberger et al., 2015) and add temporal transformers after image weights (commonly referred to as the 2+1D paradigm) as denoisers (Yuan et al., 2025c; 2024; Guo et al., 2023; Wang et al., 2025; Chen et al., 2023a; 2024a). After the release of Sora (Brooks et al., 2024), the community uses Diffusion Transformers (DiTs) (Peebles & Xie, 2023; Yao et al., 2025) as denoisers (Zheng et al., 2024; Lin et al., 2024a; Ma et al., 2024; Xu et al., 2024) within the 2+1D paradigm to achieve T2V. To overcome the limited capability of the 2+1D paradigm in temporal modeling, approaches like Open-Sora Plan v1.2 (Lin et al., 2024a) model all tokens uniformly (commonly referred to as the 3D paradigm) instead of differentiating between image and temporal weights. At present, with the adoption of 3D Transformer and more advanced diffusion models, flow matching (Lipman et al., 2022; Liu, 2022), T2V models can generate highly realistic videos.

### 2.2 IMAGE-TO-VIDEO GENERATION

Image-to-Video Generation leverages a conditional image and a prompt as inputs, enhancing the controllability of the generated video. Stable Video Diffusion (SVD) (Blattmann et al., 2023) combines conditional image latents with noisy latents, and injects high-level semantic information of the conditional image extracted by CLIP (Radford et al., 2021; Zhu et al., 2023; Lin et al., 2023; 2024b; Chen et al., 2024b) into the denoiser. DynamiCrafter (Xing et al., 2024) improves on SVD by using a query transformer (Li et al., 2023) to extract CLIP tokens. CogVideoX (Yang et al., 2024) concatenates zero-padding conditional image latents with noisy latents to introduce conditions. SEINE (Chen et al., 2023b) introduces a temporal inpainting model, where conditional image latents and mask sequences are concatenated with noisy latents to fill in subsequent frames. Open-Sora Plan v1.3 (Lin et al., 2024a; Li et al., 2025b) further expands the inpainting model to cover more tasks and proposes a progressive training strategy to enhance performance. Wan2.1 (Wan et al., 2025) improves the inpainting model by introducing semantic information extracted by CLIP. All of these approaches inject full conditional image information into the denoiser through a concatenation operation, resulting in excellent fidelity for the first frame.

### 2.3 CONDITIONAL IMAGE LEAKAGE

Conditional image leakage (Zhao et al., 2024; Yuan et al., 2025a) is an issue where the model shortcuts the conditional image information, especially at large time steps, rather than utilizing it as an auxiliary to generate the video from noisy latents. SVD introduces adding a small amount of noise to the conditional image to increase the dynamic degree, marking the first attempt to reduce conditional image leakage. Previous work (Zhao et al., 2024) proposes addressing the leakage by starting the generation process from an earlier time step during inference and designing a time-dependent noise distribution for the conditional image during training. Additionally, Adaptive Low-pass Guidance (ALG) (Choi et al., 2025) is a training-free approach by using the low-pass information of the conditional image rather than its full information at large time steps. At present, resolving conditional image leakage remains an open problem, with no universally accepted solution in the community.

## 3 METHOD

In this section, we first introduce the preliminary knowledge of flow matching in Sec. 3.1. Then, in Sec. 3.2, we present latent shifting for introducing conditions implicitly based on the characteristics of flow matching. Finally, in Sec. 3.3, we bring in Fourier guidance that injects high-frequency magnitude features extracted from the Fourier Transform into the denoiser to serve as a supplement.

### 3.1 PRELIMINARY FOR FLOW MATCHING

Continuous Normalizing Flows (CNFs) (Chen, 2018) aim to learn a transformation from a sample $z_1$ from a source distribution $q_1(z)$ to a sample $z_0$ from a target distribution $q_0(z)$, where $q_0(z)$ represents the data distribution, and $q_1(z)$ is typically a known prior distribution, such as a standard normal distribution. This transformation is usually modeled as an ordinary differential equation (ODE) with $t \in [0, 1]$. Let $z_t$ represents the intermediate state from $z_1$ to $z_0$, then the transformation is governed by the following equation:

$$\frac{dz_t}{dt} = v_t(z_t, t), t \in [0, 1]. \tag{1}$$

Here, $v_t(z_t, t)$ defines the velocity field at any time, dictating how the distribution transfers over time. The concept of Flow Matching (FM) (Lipman et al., 2022; Liu, 2022; Tong et al., 2023) is to directly learn the vector field $v_t(z_t, t)$ from $z_1$ to $z_0$ using a neural network $v_\theta(z_t, t)$. Specifically, for $z_1 \sim q_1$ and $z_0 \sim q_0$, their linear interpolation is constructed as follows:

$$z_t = (1 - t)z_0 + tz_1, t \in [0, 1]. \tag{2}$$

The vector field $v_t(z_t, t)$ in this interpolation mode is given by:

$$v_t(z_t, t) = \frac{dz_t}{dt} = z_1 - z_0. \tag{3}$$

$v_t(z_t, t)$ is only related to the two points $z_0$ and $z_1$ of the probability path and is independent of $t$. The optimization objective of FM is to train a neural network $v_\theta(z_t, t)$ to approximate $v_t(z_t, t)$ using the Mean Squared Error (MSE) loss. Under a condition $y$, flow matching can be modeled as Conditional Flow Matching (CFM). The optimization objective of CFM is:

$$\mathcal{L}_{\text{CFM}}(\theta) = \mathbb{E}_{t \sim \mathcal{U}[0,1], z_0 \sim q_0, z_1 \sim q_1} \left[ \| v_\theta((1 - t)z_0 + tz_1, t, y) - (z_1 - z_0) \|_2^2 \right], \tag{4}$$

where $\mathcal{U}[0, 1]$ represents the uniform distribution over $[0, 1]$. During sampling, we sample $z_1 \sim q_1$ and solve the ODE $\frac{dz_t}{dt} = v_\theta(z_t, t, y)$ step by step from $t = 1$ to $t = 0$ to obtain $z_0 \sim q_0$. Unlike Denoising Diffusion Probabilistic Models (DDPM) (Ho et al., 2020), which require the source distribution to be a standard normal distribution, FM imposes no such constraints on the source distribution. This flexibility allows FM to be used for transferring between any two distributions.

### 3.2 LATENT SHIFTING

We consider implementing I2V without explicitly incorporating the full information of the conditional image into the hidden states of the denoiser. Let $\epsilon \sim \mathcal{N}(0, I)$, where $\mathcal{N}(0, I)$ denotes a standard normal distribution. Let a conditional image be $S \in \mathbb{R}^{c \times h \times w}$, and a video starting with the conditional image be $X \in \mathbb{R}^{c \times t \times h \times w}$, which means $X[:, 0] = S$. Let $\mathcal{E}$ represent the encoder of the Variational Autoencoder (VAE) (Kingma & Welling, 2013). Denote the source distribution sample as $z_1^T$ and the target distribution sample as $z_0^T$ for the T2V task, we have $z_1^T = \epsilon$, $z_0^T = x = \mathcal{E}(X)$, and the intermediate state $z_t^T$ at any time $t$ under FM is given by:

$$z_t^T = (1 - t)z_0^T + tz_1^T = (1 - t)x + t\epsilon. \tag{5}$$

Let the velocity field for the T2V task be $v_t^T(z_t^T, t)$, then $v_t^T(z_t^T, t) = \frac{dz_t^T}{dt} = \epsilon - x$. For the I2V task, let the source distribution sample be $z_1^I$ and the target distribution sample be $z_0^I$. Since FM imposes no constraints on the source and target distributions, we can modify the distributions to implicitly incorporate conditions and avoid conditional image leakage. $z_1^I$ is modified to a linear

Figure 2: **Method overview**. We extract features from the conditional image latents using a learnable projection, followed by the latent shifting to obtain a renewed intermediate state that implicitly contains the condition. Simultaneously, the conditional image latents undergo the Fourier Transform to extract high-frequency magnitude features as guidance, which are concatenated with noisy latents and injected into DiT. During inference, we begin with the shifted noise and progressively denoise following the ODE, ultimately decoding the video.

mixture of the conditional image latents $s = \mathcal{E}(S)$ and noise $\epsilon$, and $z_0^I$ is modified to a linear mixture of the conditional image latents $s$ and the video $x$, as follows:

$$z_1^I = \alpha s + \beta \epsilon, \tag{6}$$

$$z_0^I = \gamma s + \kappa x, \tag{7}$$

where $\alpha, \beta, \gamma, \kappa$ are undetermined constant numbers. We can compute the intermediate state $z_t^I$ as:

$$z_t^I = (1 - t)z_0^I + tz_1^I = \kappa z_t^T + [\gamma + (\alpha - \gamma)t]s + (\beta - \kappa)t\epsilon. \tag{8}$$

For the I2V task, let the velocity field be $v_t^I(z_t^I, t)$, which can be expressed as:

$$v_t^I(z_t^I, t) = \frac{dz_t^I}{dt} = \kappa v_t^T(z_t^T, t) + (\alpha - \gamma)s + (\beta - \kappa)\epsilon. \tag{9}$$

When training an I2V model, we typically inherit the weight of the corresponding T2V model. A good initialization can leverage knowledge from the pre-trained weights as much as possible. It is observed that when $\alpha = \gamma$ and $\beta = \kappa = 1$, we have $v_t^I(z_t^I, t) = v_t^T(z_t^T, t)$, meaning the optimization objectives for I2V and T2V are structurally the same. In this case, $z_t^I$ can be expressed as:

$$z_t^I = z_t^T + \gamma s. \tag{10}$$

When $\gamma = 0$, we have $z_t^I = z_t^T$, meaning that without the conditional image as a condition, the model is equivalent to a T2V model. When $\gamma \neq 0$, the input of the denoiser incorporates conditional image information. In this case, we have $z_1^I = \epsilon - (-\gamma s)$ and $z_0^I = x - (-\gamma s)$, where $-\gamma s$ can be viewed as the shifting of the latents in I2V task relative to T2V task and we aim to learn the conditional image information from the shifted latents.

Furthermore, $-\gamma$ can be viewed as a constant weight for $s$. Since the effective information of each position varies, we can replace $-\gamma s$ with $\phi(s)$, where $\phi(\cdot)$ is a learnable projection. Additionally, we can zero-initialize $\phi(\cdot)$, ensuring that the input distribution of the denoiser is not disrupted at the start of training. Since $\phi(\cdot)$ is a network that is independent of time $t$, we still have $v_t^I(z_t^I, t) = v_t^T(z_t^T, t)$. Now we obtain the ultimate form for the I2V method based on the latent shifting:

$$z_1^I = \epsilon - \phi(s), \tag{11}$$

$$z_0^I = x - \phi(s), \tag{12}$$

$$z_t^I = z_t^T - \phi(s) = (1 - t)x + t\epsilon - \phi(s), \tag{13}$$

$$v_t^I(z_t^I, t) = v_t^T(z_t^T, t) = \epsilon - x. \tag{14}$$

### 3.3 FOURIER GUIDANCE

During sampling, the latent shifting method needs to recover the information of $s$ from the mixture of $\epsilon$ and $\phi(s)$. While recovery of low-frequency information like global color and shape is easier, high-frequency details like edges and contours are more challenging to recover accurately from the shifted noise. Therefore, models trained with the latent shifting require more time and data than the existing I2V paradigms to ensure the fidelity for the first frame.

We consider injecting the high-frequency information of $s$ as additional input, aiming to address the challenge of learning high-frequency information. Since VAEs in latent diffusion models function similarly to AutoEncoders (AEs), we find that low-frequency and high-frequency information from the Fourier Transform in latent space resembles that in pixel space, but with significantly lower computational cost, as shown in the App. C. Since directly using high-frequency features leads to shortcutting by the model, we only retain the magnitudes of these features. Let $\boldsymbol{f}_{\text{high}}$ be the high-frequency magnitude filter of the Fourier Transform, we have:

$$\boldsymbol{s}_{\text{high}} = \boldsymbol{f}_{\text{high}}(\boldsymbol{s}), \tag{15}$$

where $\boldsymbol{s}_{\text{high}}$ means high-frequency magnitude features of $\boldsymbol{s}$. The detailed implementation of $\boldsymbol{f}_{\text{high}}$ can be found in the App. C. Then, $\boldsymbol{s}_{\text{high}}$ is concatenated with $\boldsymbol{z}_t^I$ along the channel dimension. After forwarding the embedding layers, we obtain the hidden states of the denoiser $\boldsymbol{H}$:

$$\boldsymbol{H} = \begin{bmatrix} \boldsymbol{W}^I & \boldsymbol{W}^F \end{bmatrix} \begin{bmatrix} \boldsymbol{z}_t^I \\ \boldsymbol{s}_{\text{high}} \end{bmatrix}, \tag{16}$$

where $[\cdot]$ denotes concatenation along channel dimension. Here, $\boldsymbol{W}^I$ represents patch embedding of the denoiser, and $\boldsymbol{W}^F$ is the embedding layer corresponding to $\boldsymbol{s}_{\text{high}}$. $\boldsymbol{W}^F$ is zero-initialized to ensure that the distribution of the hidden states remains unchanged at the beginning of training.

In summary, we can derive the loss function implemented by FlashI2V as follows:

$$\mathcal{L}_{\text{Flash}}(\theta) = \mathbb{E}_{t \sim \mathcal{U}[0,1], \epsilon \sim \mathcal{N}(\boldsymbol{0}, \boldsymbol{I}), \boldsymbol{X} \sim q(\boldsymbol{X}), \boldsymbol{x} = \mathcal{E}(\boldsymbol{X}), \boldsymbol{s} = \mathcal{E}(\boldsymbol{X}[:,0])} \left[ \left\| \boldsymbol{v}_\theta^I((1-t)\boldsymbol{x} + t\epsilon - \phi(\boldsymbol{s}), t, \boldsymbol{y}, \boldsymbol{s}_{\text{high}}) - (\epsilon - \boldsymbol{x}) \right\|_2^2 \right], \tag{17}$$

where $\boldsymbol{y}$ is the text embedding, and $\boldsymbol{v}_\theta^I$ is the denoiser excluding $\phi$.

## 4 EXPERIMENT

In this section, we first introduce the experimental setup in Sec. 4.1. Then, in Sec. 4.2, we compare FlashI2V with other I2V methods from both quantitative and qualitative perspectives. In addition, in Sec. 4.3, we present the results of the ablation experiments to demonstrate the effectiveness of FlashI2V. Finally, in Sec. 4.4, we analyze the functions of the modules in FlashI2V.

### 4.1 EXPERIMENTAL SETUP

**Training Setup**. In the comparisons, we train a model for 84K steps on 20M high-quality video data collected internally, following the collection and processing pipeline described in Open-Sora Plan (Lin et al., 2024a). For each video, we randomly sample 49 frames at a fixed fps of 16, with a resolution of $480 \times 832$. We initialize the model from the Wan2.1-T2V-1.3B (Wan et al., 2025) model. The learnable projection is implemented using two layers of Conv3D (Tran et al., 2015) and SiLU (Elfwing et al., 2018), and the Fourier embedding layer is implemented in the same way as the patch embedding, both with zero initialization. During training, the first frame of each video serves as the conditional image. The cutoff frequency percentile of the Fourier Transform is sampled from $\mathcal{U}[0.05, 0.95]$. The text prompt is dropped with a probability of 0.1. We use a batch size of 256, a learning rate of 4e-5, a weight decay of 1e-2, and the AdamW optimizer with $\beta_1$ set to 0.9, $\beta_2$ set to 0.999, and $\epsilon$ set to 1e-15. The weights are updated using Exponential Moving Average (EMA) with a decay of 0.9999. In the ablation study, all models involved in the comparison are initialized from Wan2.1-T2V-1.3B. We select a 2M subset as the training set, use a learning rate of 2e-5, a batch size of 64, and 30K training steps, while keeping the other settings unchanged.
**Sampling Setup**. For sampling, we use the Discrete Euler Sampler with a sigma shifting strategy as in HunyuanVideo (Kong et al., 2024), a shifting coefficient of 7.0, classifier-free guidance set to 5.0, 50 sampling steps, and a cutoff frequency percentile set to 0.1.

Table 1: **Vbench-I2V results**. We compare the performance of various methods on Vbench-I2V. It can be observed that, despite having the fewest parameters, FlashI2V achieves comparable scores to models with larger parameter sizes. The dynamic degree score of FlashI2V significantly surpasses that of other methods. All scores are presented as percentages (%). † indicates testing using recaptioning image-text pairs on Vbench-I2V. For further details, see the App. B.

| Model | I2V Paradigm | Subject Consistency↑ | Background Consistency↑ | Motion Smoothness↑ | Dynamic Degree↑ | Aesthetic Quality↑ | Imaging Quality↑ | I2V Subject Consistency↑ | I2V Background Consistency↑ |
|---|---|---|---|---|---|---|---|---|---|
| SVD-XT-1.0 (1.5B) | Repeating Concat and Adding Noise | 95.52 | 96.61 | 98.09 | 52.36 | 60.15 | 69.80 | 97.52 | 97.63 |
| SVD-XT-1.1 (1.5B) | Repeating Concat and Adding Noise | 95.42 | 96.77 | 98.12 | 43.17 | 60.23 | 70.23 | 97.51 | 97.62 |
| SEINE-512x512 (1.8B) | Inpainting | 95.28 | 97.12 | 97.12 | 27.07 | 64.55 | **71.39** | 97.15 | 96.94 |
| CogVideoX-5B-I2V | Zero-padding Concat and Adding Noise | 94.34 | 96.42 | 98.40 | 33.17 | 61.87 | 70.01 | 97.19 | 96.74 |
| Wan2.1-I2V-14B-720P | Inpainting | 94.86 | 97.07 | 97.90 | 51.38 | 64.75 | 70.44 | 96.95 | 96.44 |
| CogVideoX1.5-5B-I2V† | Zero-padding Concat and Adding Noise | 95.04 | 96.52 | **98.47** | 37.48 | **62.68** | 70.99 | 97.78 | 98.73 |
| Wan2.1-I2V-14B-480P† | Inpainting | **95.68** | **97.44** | 98.46 | 45.20 | 61.44 | 70.37 | **97.83** | **99.08** |
| **FlashI2V† (1.3B)** | **FlashI2V** | 95.13 | 96.36 | 98.35 | **53.01** | 62.34 | 69.41 | 97.67 | 98.72 |

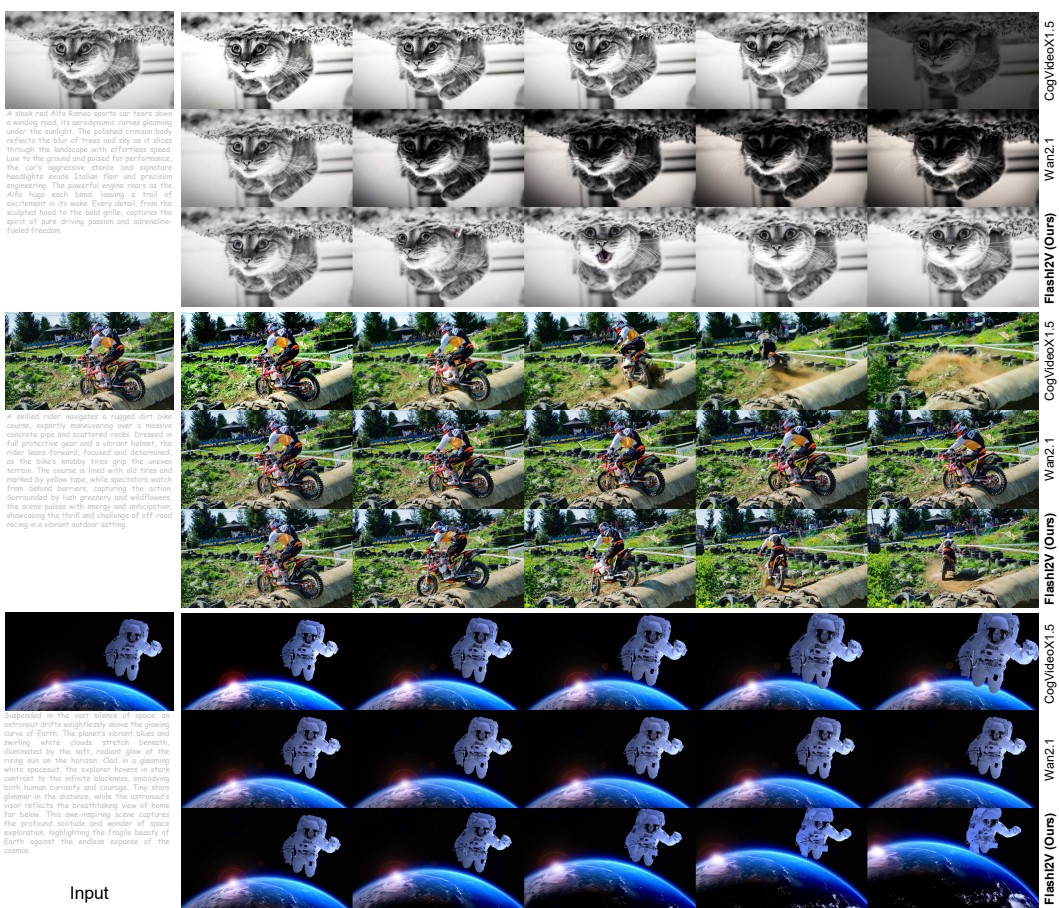

Figure 3: **Method Comparison**. We compare the quantitative performance of FlashI2V (1.3B) with CogVideoX1.5-5B-I2V (Yang et al., 2024) and Wan2.1-I2V-14B-480P (Wan et al., 2025). We observe that CogVideoX1.5 and Wan2.1 exhibit color inconsistency. Additionally, Wan2.1 tends to produce extremely slow-motion or even static videos. Thanks to the avoidance of conditional image leakage, FlashI2V effectively resolves these performance degradation issues.

**Evaluation**. In the comparisons, we use a fixed 49 frames and the default resolution, and we utilize all Vbench-I2V (Huang et al., 2024; Zheng et al., 2025) metrics except for Camera Motion as evaluation metrics. In addition, we use ChatGPT (Achiam et al., 2023) to rewrite the short prompts of Vbench-I2V in order to obtain more accurate evaluation results. For further details, please refer to the App. B. In the ablation study, we randomly select 1,000 videos from the HD subset of OpenVid-1M (Nan et al., 2024) as the validation set, calculating the chunk-wise FVD for each setting.

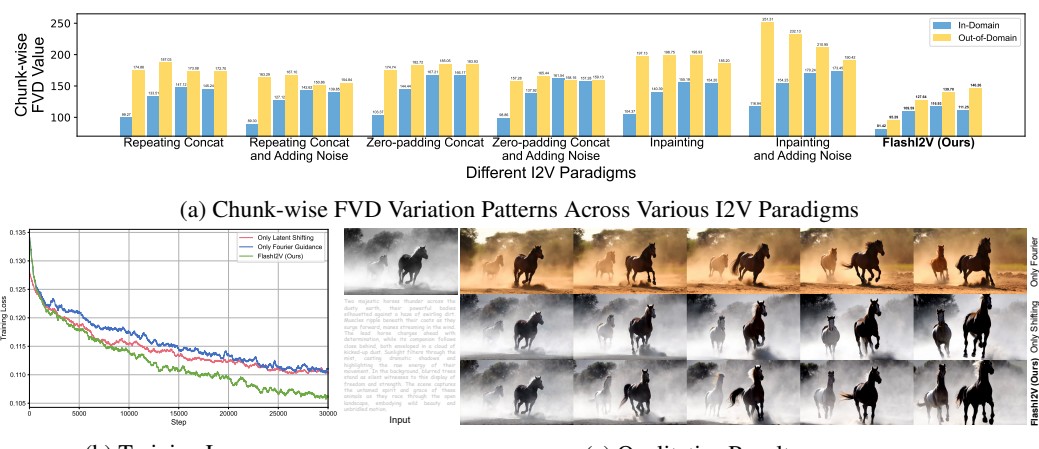

(a) Chunk-wise FVD Variation Patterns Across Various I2V Paradigms

(b) Training Loss

(c) Qualitative Results

Figure 4: **Ablation Study**. (a) Comparing the chunk-wise FVD variation patterns of different I2V paradigms on both the training and validation sets, it is observed that only FlashI2V exhibits the same time-increasing FVD variation pattern in both sets. This suggests that only FlashI2V is capable of applying the generation law learned from in-domain data to out-of-domain data. Additionally, FlashI2V has the lowest out-of-domain FVD, demonstrating its performance advantage. (b) From the training loss, we can observe that Fourier guidance accelerates the convergence of latent shifting. (c) Fourier guidance alone causes color deviation, while latent shifting alone leads to mismatched details. FlashI2V achieves consistency in both color and details.

## 4.2 MAIN RESULTS

**Quantitative results.** We compare the performance of different methods on Vbench-I2V, as shown in Tab. 1. Because of preventing conditional image leakage, FlashI2V achieves a significantly higher dynamic degree score across all methods. In other metrics, FlashI2V is quite close to CogVideoX1.5-5B-I2V and Wan2.1-I2V-14B-480P with larger parameter sizes. It outperforms CogVideoX1.5 in the Subject Consistency metric and exceeds Wan2.1 in the Aesthetic Quality metric.

**Qualitative results.** As shown in Fig. 3, we compare the qualitative performance of different methods. Due to the impact of conditional image leakage, CogVideoX1.5 and Wan2.1 exhibit issues such as color inconsistency and slow motion in some samples. Wan2.1 even produces completely static videos, which can be referred to in the supplementary materials. In contrast, FlashI2V generates videos with larger motion and adheres more closely to physical laws.

## 4.3 ABLATION STUDY

**Generalization to out-of-domain data**. We compare the chunk-wise FVD variation patterns of different I2V paradigms on in-domain and out-of-domain data, as shown in Fig. 4a. The pseudocode implementations of the different paradigms can be found in the App. D. Each generated video is divided into four temporal chunks over time with an equal interval. Apart from FlashI2V, other paradigms concatenate the full information of the conditional image with the noisy latents. The paradigms named with "Adding Noise" add a small amount of noise to the conditional image latents, similar to the implementation in CogVideoX (Yang et al., 2024). We observe that only FlashI2V exhibits the same chunk-wise FVD variation pattern on both in-domain and out-of-domain data, indicating that the generation law learned by FlashI2V on in-domain data generalizes well to out-of-domain data. In contrast, other paradigms show inconsistent chunk-wise FVD variation patterns between in-domain and out-of-domain data, suggesting the leakage caused by shortcutting conditional images. Moreover, our method achieves the lowest FVD on out-of-domain data, meaning it has the best performance across different I2V paradigms.

**The functions of various modules in FlashI2V**. To investigate the effectiveness of latent shifting and Fourier guidance, we conduct detailed ablation experiments. From a quantitative perspective, as shown in Fig. 4b, FlashI2V achieves a faster decline in training loss by incorporating Fourier guidance compared to using latent shifting alone, indicating that Fourier guidance effectively accelerates the

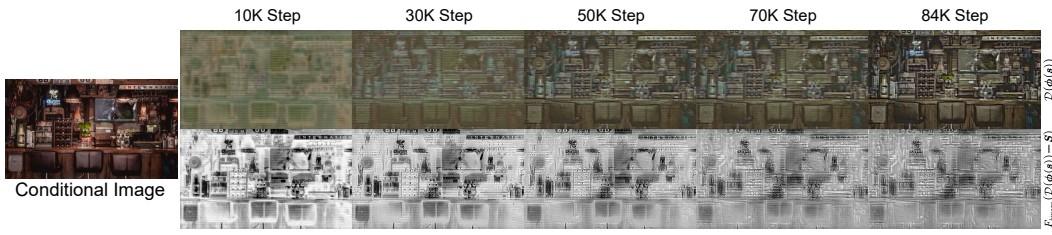

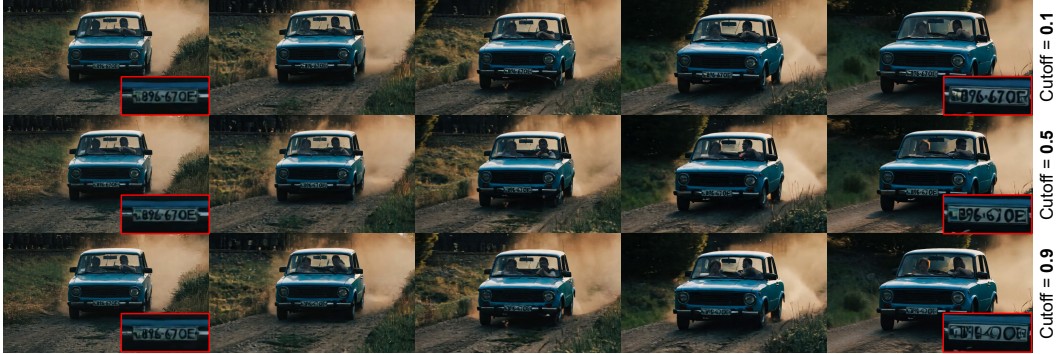

(a) Encoded Features of Latent Shifting

(b) The Influence of Fourier Guidance on Generation Details

Figure 5: **Analysis of latent shifting and fourier guidance**. (a) As training progresses, $\phi(\cdot)$ gradually emphasizes the detailed information in the conditional image. (b) When a lower cutoff frequency percentile is used, more high-frequency information is injected. When the cutoff frequency percentile is set to 0.1, the graphical text at the end of the video remains unchanged, while with the cutoff frequency percentile set to 0.9, the graphical text becomes unrecognizable.

convergence of latent shifting. From a qualitative perspective, we compare the performance after removing different modules in Fig. 4c. When using only Fourier guidance, the generated video maintains high-frequency content consistent with the conditional image. However, due to receiving only the magnitude features, it fails to produce correct colors with only Fourier guidance. With only latent shifting, the generated scene aligns with the conditional image but lacks satisfactory fidelity in local details. FlashI2V successfully achieves both global and local fidelity.

## 4.4 ANALYSIS

**Features encoded through latent shifting**. Since $\phi(\cdot)$ performs a shifting operation on latents, its encoded features are meaningful in the latent space. As shown in Fig. 5a, we visualize the features encoded by $\phi(\cdot)$ in the pixel space by VAE decoding and compute the relative difference between $\mathcal{D}(\phi(s))$ and $S$ in the pixel space, represented as a binary image. As training progresses, the features encoded by $\phi(\cdot)$ become richer, and $\phi(s)$ emphasizes high-frequency representations compared to $s$, resulting in gradually improved fidelity of the model during training.

**Adjustable generation details**. At different cutoff frequency percentiles, Fourier guidance can provide varying detail levels. As shown in Fig. 5b, we compare the influence of changing cutoff frequency percentiles on the generated details. A lower cutoff frequency percentile means injecting richer high-frequency details, resulting in finer generation and better detail preservation during the entire video, especially for small-scale regions like graphical text.

## 5 CONCLUSION

Existing I2V paradigms cannot avoid conditional image leakage, leading to performance degradation. We propose FlashI2V, which implicitly introduces conditions through latent shifting. Additionally, we utilize high-frequency magnitude features extracted by the Fourier Transform as guidance to accelerate the convergence. Experimental results show that FlashI2V demonstrates the best generalization and performance on out-of-domain data. With only 1.3B parameters, FlashI2V achieves the best dynamic degree score across various methods on Vbench-I2V.

## 6 REPRODUCIBILITY STATEMENT

In this work, all methods, experiments, and evaluation metrics are reproducible, as explained below:

- We provide the pseudocode implementations of all I2V paradigms in the App. D, and all I2V paradigms are trained on the same dataset using the same random seed.
- We provide the complete training and inference code for FlashI2V. For more details, please refer to the supplementary materials.
- For the recaptioning of text-image pairs in Vbench-I2V, we provide the specific implementation in the App. B, and the refined prompts are included in the supplementary materials.
- The training data sources are detailed in the App. F. In addition, as described in Sec. 4.1, the data collection and processing pipeline follows the Open-Sora Plan (Lin et al., 2024a).
- The code used to calculate FVD, the model sources used in Vbench-I2V, and the source links for CogVideoX1.5 and Wan2.1 are provided in the App. G.

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

# FLASHI2V: FOURIER-GUIDED LATENT SHIFTING PREVENTS CONDITIONAL IMAGE LEAKAGE IN IMAGE-TO-VIDEO GENERATION
## APPENDIX

## A  THE USE OF LARGE LANGUAGE MODELS (LLMs)

This work utilizes various LLMs, and their roles are as follows:

- **Polishing tool**. The authors utilize the GPT-4o as a tool to polish and refine the writing.

- **Recaptioning for text-image pairs in Vbench-I2V**. Since the texts of the text-video pairs in the training set consist solely of long texts, and the prompts used in Vbench-I2V (Huang et al., 2024; Zheng et al., 2025) are short, we use the gpt-4.1-2025-04-14 API to perform recaptioning on each image in Vbench-I2V based on its provided short prompt. Detailed information is available in the supplementary materials.

- **Captioning for videos in training data**. Following Open-Sora Plan (Lin et al., 2024a), we annotate our data with QWen2-VL-7B (Wang et al., 2024) using the same prompt.

## B  RECAPTIONING FOR TEXT-IMAGE PAIRS IN VBENCH-I2V

FlashI2V is initialized from the Wan2.1-T2V-1.3B weights. Since Wan2.1 is trained on text-video pairs with long captions, and our training set also consists of such pairs, we perform recaptioning for the text-image pairs of Vbench-I2V to accurately evaluate model performance.

We use the GPT-4.1-2025-04-14 API for recaptioning, with the following prompt design:

```
You are a professional text editor, skilled at optimizing video
descriptions.  You will be given an image and a text description
of the video's content, starting with the input image.
Please polish the input description to make it more vivid,
concise, and expressive, while preserving the original meaning.
Please limit the polished description to between 100-150 words.
The new prompt directly describes the specific content without
words such as video, image, or picture.
description:  {caption}.
```

As shown in Fig. 6, the original short prompt is refined into a long prompt, adding more detailed descriptions while maintaining the original meaning.

## C  HIGH-FREQUENCY MAGNITUDE FEATURES IN FOURIER GUIDANCE

In Sec. 3.3, we discuss using the high-frequency magnitude features extracted through the Fourier transform as guidance to accelerate convergence. The complete derivation is as follows:

Let $\mathbf{FFT}(\cdot)$ denote the Fourier Transform and $\mathbf{iFFT}(\cdot)$ denote the inverse Fourier Transform. We first apply $\mathbf{FFT}(\cdot)$ to the conditional image latents $\boldsymbol{s}$ to obtain the frequency spectrum of $\boldsymbol{s}$:

$$\mathbf{FFT}(\boldsymbol{s}) \triangleq \hat{\boldsymbol{s}}_c^{\text{freq}}(u, v)$$
$$= \sum_{h=0}^{H-1} \sum_{w=0}^{W-1} \boldsymbol{s}_c(h, w) \exp\left(-2\pi i \left(\frac{uh}{H} + \frac{vw}{W}\right)\right), \tag{18}$$

where $\boldsymbol{s}_c(h, w)$ is the value in the c-th channel in the spatial domain and $\hat{\boldsymbol{s}}_c^{\text{freq}}(u, v)$ is the complex frequency data in the frequency domain. We remove the phase information from the frequency spectrum and retain only the magnitude, resulting in the magnitude map $\boldsymbol{M}_c(u, v)$:

$$\boldsymbol{M}_c(u, v) = |\hat{\boldsymbol{s}}_c^{\text{freq}}(u, v)| = \sqrt{\Re(\hat{\boldsymbol{s}}_c^{\text{freq}}(u, v))^2 + \Im(\hat{\boldsymbol{s}}_c^{\text{freq}}(u, v))^2}, \tag{19}$$

a bald eagle flying over a tree filled forest.
Original Prompt

A majestic bald eagle soars gracefully above a lush, tree-filled forest, its powerful wings stretched wide as it glides effortlessly through the air. Sunlight catches the eagle's striking white head and tail, contrasting sharply with its dark, sleek feathers. The vibrant green canopy below provides a vivid backdrop, highlighting the bird's commanding presence and keen focus. With sharp eyes scanning the landscape, the eagle embodies freedom and strength, moving silently over the treetops in search of prey or a place to perch. The scene captures the raw beauty and untamed spirit of the wilderness, showcasing the bald eagle in its natural element, ruling the skies with elegance and authority.

Image

Refined Prompt

Figure 6: **Racaptioning for text-image pairs in Vbench-I2V**. With recaptioning, the refined prompt now includes more detailed descriptions, aligning better with the distribution of the training set, which enhances the inference performance of the model to a more realistic result.

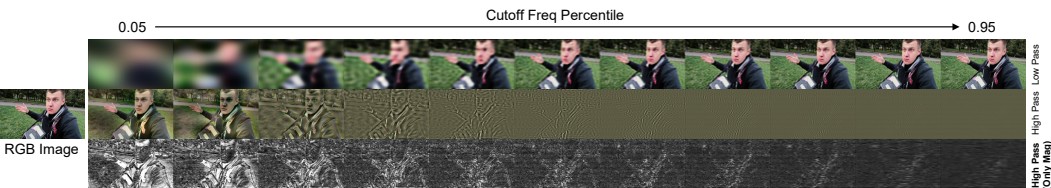

Figure 7: **Information extracted by the Fourier Transform**. After performing the Fourier Transform in the latent space and decoding features to the pixel space, we observe that as the cutoff frequency percentile increases, the high-frequency information in the latents diminishes. We extract only the magnitude of the high-frequency information, ensuring that the original high-frequency information cannot be restored while still providing guidance.

where $\Re(\hat{s}_c^{\text{freq}}(u,v))$ and $\Im(\hat{s}_c^{\text{freq}}(u,v))$ represent the real and imaginary parts of the complex frequency, respectively. Then, a specified cutoff frequency $\text{cutoff\_freq}$ is defined as:

$$\text{cutoff\_freq} = \min\{r \mid \frac{\sum_{r(u,v)\leq r} \boldsymbol{M}(u,v)}{\sum_{u,v} \boldsymbol{M}(u,v)} \geq p\}, \quad (20)$$

where $p$ is the cutoff frequency percentile, representing the percentile of low-frequency energy. In addition, $r(u,v)$ represents the radius, which can be calculated using the following formula:

$$r(u,v) = \sqrt{(u-u_0)^2 + (v-v_0)^2}, \quad (21)$$

where $(u_0, v_0)$ is the center of the frequency plane. To implement high-frequency information extraction, we define the frequency masks:

$$\boldsymbol{\text{Mask}}^{\text{low}}(u,v) = \begin{cases} 1, & \text{if } r(u,v) < \text{cutoff\_freq}, \\ 0, & \text{otherwise}, \end{cases} \quad (22)$$

$$\boldsymbol{\text{Mask}}^{\text{high}}(u,v) = 1 - \boldsymbol{\text{Mask}}^{\text{low}}(u,v).$$

After performing filtering in the frequency domain using the frequency mask $\boldsymbol{\text{Mask}}^{\text{high}}$, we can apply the inverse Fourier transform $\boldsymbol{\text{iFFT}}(\cdot)$ to obtain the high-frequency component in the spatial domain $s_c^{\text{high}}$:

$$s_c^{\text{high}} = \boldsymbol{\text{iFFT}}\left(\hat{s}_c^{\text{freq}} \cdot \boldsymbol{\text{Mask}}^{\text{high}}\right). \quad (23)$$

After performing the inverse Fourier Transform to obtain the high-frequency component $s_c^{\text{high}}$, we perform magnitude extraction to obtain the final magnitude of the high-frequency component $\boldsymbol{M}_c^{\text{high}}$:

$$\boldsymbol{M}_c^{\text{high}} = |s_c^{\text{high}}|. \quad (24)$$

The $s_{\text{high}}$ in each channel from Sec. 3.3 corresponds to $\boldsymbol{M}_c^{\text{high}}$ here.

As shown in Fig. 7, in the latent space, the amount of high-frequency information decreases as the cutoff frequency percentile increases, similar to the behavior in pixel space. Therefore, for a

---

**Algorithm 1** Sampling process for Repeating Concat paradigm

---

**Require:** Denoiser $\boldsymbol{v}_\theta$, VAE encoder $\mathcal{E}$, VAE decoder $\mathcal{D}$, input conditional image $\boldsymbol{S}$, prompt $\boldsymbol{y}$, guidance $w$, total inference steps $N$
1: $\boldsymbol{s} \leftarrow \mathcal{E}(\boldsymbol{S})$
2: $\boldsymbol{s} \leftarrow \texttt{Repeat}(\boldsymbol{s})$
3: $\boldsymbol{z} \sim \mathcal{N}(\boldsymbol{0}, \boldsymbol{I})$
4: **for** $i = 0$ to $N - 1$ **do**
5:     $t \leftarrow \frac{N-i}{N}$
6:     $\hat{\boldsymbol{z}} \leftarrow \texttt{Concat}(\boldsymbol{s}, \boldsymbol{z})$
7:     $\boldsymbol{v} \leftarrow \boldsymbol{v}_\theta(\hat{\boldsymbol{z}}, t, \varnothing) + w[\boldsymbol{v}_\theta(\hat{\boldsymbol{z}}, t, \boldsymbol{y}) - \boldsymbol{v}_\theta(\hat{\boldsymbol{z}}, t, \varnothing)]$
8:     $\boldsymbol{z} \leftarrow \texttt{SolverStep}(\boldsymbol{z}, \boldsymbol{v}, t)$
9: **end for**
10: **return** $\mathcal{D}(\boldsymbol{z})$

---

**Algorithm 2** Sampling process for Repeating Concat and Adding Noise paradigm

---

**Require:** Denoiser $\boldsymbol{v}_\theta$, VAE encoder $\mathcal{E}$, VAE decoder $\mathcal{D}$, input conditional image $\boldsymbol{S}$, prompt $\boldsymbol{y}$, guidance $w$, total inference steps $N$, mean of adding noise $\boldsymbol{\mu}$, variance of adding noise $\boldsymbol{\sigma^2}$
1: $\boldsymbol{\epsilon} \sim \mathcal{N}(\boldsymbol{\mu}, \boldsymbol{\sigma^2})$
2: $\boldsymbol{S} \leftarrow \texttt{Add\_Noise}(\boldsymbol{S}, \boldsymbol{\epsilon})$
3: $\boldsymbol{s} \leftarrow \mathcal{E}(\boldsymbol{S})$
4: $\boldsymbol{s} \leftarrow \texttt{Repeat}(\boldsymbol{s})$
5: $\boldsymbol{z} \sim \mathcal{N}(\boldsymbol{0}, \boldsymbol{I})$
6: **for** $i = 0$ to $N - 1$ **do**
7:     $t \leftarrow \frac{N-i}{N}$
8:     $\hat{\boldsymbol{z}} \leftarrow \texttt{Concat}(\boldsymbol{s}, \boldsymbol{z})$
9:     $\boldsymbol{v} \leftarrow \boldsymbol{v}_\theta(\hat{\boldsymbol{z}}, t, \varnothing) + w[\boldsymbol{v}_\theta(\hat{\boldsymbol{z}}, t, \boldsymbol{y}) - \boldsymbol{v}_\theta(\hat{\boldsymbol{z}}, t, \varnothing)]$
10:     $\boldsymbol{z} \leftarrow \texttt{SolverStep}(\boldsymbol{z}, \boldsymbol{v}, t)$
11: **end for**
12: **return** $\mathcal{D}(\boldsymbol{z})$

---

**Algorithm 3** Sampling process for Zero-Padding Concat paradigm

---

**Require:** Denoiser $\boldsymbol{v}_\theta$, VAE encoder $\mathcal{E}$, VAE decoder $\mathcal{D}$, input conditional image $\boldsymbol{S}$, prompt $\boldsymbol{y}$, guidance $w$, total inference steps $N$
1: $\boldsymbol{s} \leftarrow \mathcal{E}(\boldsymbol{S})$
2: $\boldsymbol{s} \leftarrow \texttt{Pad\_Zeros}(\boldsymbol{s})$
3: $\boldsymbol{z} \sim \mathcal{N}(\boldsymbol{0}, \boldsymbol{I})$
4: **for** $i = 0$ to $N - 1$ **do**
5:     $t \leftarrow \frac{N-i}{N}$
6:     $\hat{\boldsymbol{z}} \leftarrow \texttt{Concat}(\boldsymbol{s}, \boldsymbol{z})$
7:     $\boldsymbol{v} \leftarrow \boldsymbol{v}_\theta(\hat{\boldsymbol{z}}, t, \varnothing) + w[\boldsymbol{v}_\theta(\hat{\boldsymbol{z}}, t, \boldsymbol{y}) - \boldsymbol{v}_\theta(\hat{\boldsymbol{z}}, t, \varnothing)]$
8:     $\boldsymbol{z} \leftarrow \texttt{SolverStep}(\boldsymbol{z}, \boldsymbol{v}, t)$
9: **end for**
10: **return** $\mathcal{D}(\boldsymbol{z})$

---

conditional image, we apply the above operation in the latent space to save computational resources. If we use the original extracted high-frequency features, the resulting features still contain information such as color, which can be easily shortcut. By retaining only the magnitude, we preserve the relative strength of the signal, thus emphasizing the role of guidance without a shortcut.

## D  PSEUDO-CODE IMPLEMENTATION OF DIFFERENT I2V PARADIGMS

In Sec. 4.3, we compare the chunk-wise FVD of FlashI2V with existing I2V paradigms. The pseudo-code implementations for the sampling process of various paradigms are as follows.

---

**Algorithm 4** Sampling process for Zero-Padding Concat and Adding Noise paradigm

---

**Require:** Denoiser $\boldsymbol{v}_\theta$, VAE encoder $\mathcal{E}$, VAE decoder $\mathcal{D}$, input conditional image $\boldsymbol{S}$, prompt $\boldsymbol{y}$, guidance $w$, total inference steps $N$, mean of adding noise $\boldsymbol{\mu}$, variance of adding noise $\boldsymbol{\sigma}^2$
1: $\boldsymbol{\epsilon} \sim \mathcal{N}(\boldsymbol{\mu}, \boldsymbol{\sigma}^2)$
2: $\boldsymbol{S} \leftarrow \texttt{Add\_Noise}(\boldsymbol{S}, \boldsymbol{\epsilon})$
3: $\boldsymbol{s} \leftarrow \mathcal{E}(\boldsymbol{S})$
4: $\boldsymbol{s} \leftarrow \texttt{Pad\_Zeros}(\boldsymbol{s})$
5: $\boldsymbol{z} \sim \mathcal{N}(\boldsymbol{0}, \boldsymbol{I})$
6: **for** $i = 0$ to $N - 1$ **do**
7: $\quad t \leftarrow \frac{N-i}{N}$
8: $\quad \hat{\boldsymbol{z}} \leftarrow \texttt{Concat}(\boldsymbol{s}, \boldsymbol{z})$
9: $\quad \boldsymbol{v} \leftarrow \boldsymbol{v}_\theta(\hat{\boldsymbol{z}}, t, \varnothing) + w[\boldsymbol{v}_\theta(\hat{\boldsymbol{z}}, t, \boldsymbol{y}) - \boldsymbol{v}_\theta(\hat{\boldsymbol{z}}, t, \varnothing)]$
10: $\quad \boldsymbol{z} \leftarrow \texttt{SolverStep}(\boldsymbol{z}, \boldsymbol{v}, t)$
11: **end for**
12: **return** $\mathcal{D}(\boldsymbol{z})$

---

**Algorithm 5** Sampling process for Inpainting paradigm

---

**Require:** Denoiser $\boldsymbol{v}_\theta$, VAE encoder $\mathcal{E}$, VAE decoder $\mathcal{D}$, input conditional image $\boldsymbol{S}$, prompt $\boldsymbol{y}$, guidance $w$, total inference steps $N$
1: $\boldsymbol{S} \leftarrow \texttt{Pad\_Zeros}(\boldsymbol{S})$
2: $\mathbf{M} \leftarrow \texttt{Generate\_Mask}(\boldsymbol{S})$
3: $\mathbf{m} \leftarrow \texttt{Downsample\_And\_Rearrange}(\mathbf{M})$
4: $\boldsymbol{s} \leftarrow \mathcal{E}(\boldsymbol{S})$
5: $\boldsymbol{z} \sim \mathcal{N}(\boldsymbol{0}, \boldsymbol{I})$
6: **for** $i = 0$ to $N - 1$ **do**
7: $\quad t \leftarrow \frac{N-i}{N}$
8: $\quad \hat{\boldsymbol{z}} \leftarrow \texttt{Concat}(\mathbf{m}, \boldsymbol{s}, \boldsymbol{z})$
9: $\quad \boldsymbol{v} \leftarrow \boldsymbol{v}_\theta(\hat{\boldsymbol{z}}, t, \varnothing) + w[\boldsymbol{v}_\theta(\hat{\boldsymbol{z}}, t, \boldsymbol{y}) - \boldsymbol{v}_\theta(\hat{\boldsymbol{z}}, t, \varnothing)]$
10: $\quad \boldsymbol{z} \leftarrow \texttt{SolverStep}(\boldsymbol{z}, \boldsymbol{v}, t)$
11: **end for**
12: **return** $\mathcal{D}(\boldsymbol{z})$

---

**Algorithm 6** Sampling process for Inpainting and Adding Noise paradigm

---

**Require:** Denoiser $\boldsymbol{v}_\theta$, VAE encoder $\mathcal{E}$, VAE decoder $\mathcal{D}$, input conditional image $\boldsymbol{S}$, prompt $\boldsymbol{y}$, guidance $w$, total inference steps $N$, mean of adding noise $\boldsymbol{\mu}$, variance of adding noise $\boldsymbol{\sigma}^2$
1: $\boldsymbol{\epsilon} \sim \mathcal{N}(\boldsymbol{\mu}, \boldsymbol{\sigma}^2)$
2: $\boldsymbol{S} \leftarrow \texttt{Add\_Noise}(\boldsymbol{S}, \boldsymbol{\epsilon})$
3: $\boldsymbol{S} \leftarrow \texttt{Pad\_Zeros}(\boldsymbol{S})$
4: $\mathbf{M} \leftarrow \texttt{Generate\_Mask}(\boldsymbol{S})$
5: $\mathbf{m} \leftarrow \texttt{Downsample\_And\_Rearrange}(\mathbf{M})$
6: $\boldsymbol{s} \leftarrow \mathcal{E}(\boldsymbol{S})$
7: $\boldsymbol{z} \sim \mathcal{N}(\boldsymbol{0}, \boldsymbol{I})$
8: **for** $i = 0$ to $N - 1$ **do**
9: $\quad t \leftarrow \frac{N-i}{N}$
10: $\quad \hat{\boldsymbol{z}} \leftarrow \texttt{Concat}(\mathbf{m}, \boldsymbol{s}, \boldsymbol{z})$
11: $\quad \boldsymbol{v} \leftarrow \boldsymbol{v}_\theta(\hat{\boldsymbol{z}}, t, \varnothing) + w[\boldsymbol{v}_\theta(\hat{\boldsymbol{z}}, t, \boldsymbol{y}) - \boldsymbol{v}_\theta(\hat{\boldsymbol{z}}, t, \varnothing)]$
12: $\quad \boldsymbol{z} \leftarrow \texttt{SolverStep}(\boldsymbol{z}, \boldsymbol{v}, t)$
13: **end for**
14: **return** $\mathcal{D}(\boldsymbol{z})$

---

**Repeating Concat**. As illustrated in Algorithm 1, this paradigm first repeats the conditional image latents along the temporal dimension, and then concatenates the repeated condition with noisy latents along the channel dimension to achieve I2V.

---

**Algorithm 7** Sampling process for FlashI2V paradigm

---

**Require:** Denoiser $v_\theta$, learnable projection $\phi$, VAE encoder $\mathcal{E}$, VAE decoder $\mathcal{D}$, input conditional image $S$, prompt $y$, guidance $w$, total inference steps $N$.
1: $s \leftarrow \mathcal{E}(S)$
2: $s_{\text{high}} \leftarrow \texttt{Fourier\_Filter}(s)$
3: $z \sim \mathcal{N}(\mathbf{0}, \mathbf{I})$
4: **for** $i = 0$ to $N - 1$ **do**
5:   $t \leftarrow \frac{N-i}{N}$
6:   $\hat{z} \leftarrow \texttt{Concat}(s_{\text{high}}, z - \phi(s))$
7:   $v \leftarrow v_\theta(\hat{z}, t, \varnothing) + w[v_\theta(\hat{z}_t, t, y) - v_\theta(\hat{z}_t, t, \varnothing)]$
8:   $z \leftarrow \texttt{SolverStep}(z, v, t)$
9: **end for**
10: **return** $\mathcal{D}(z)$

---

**Repeating Concat and Adding Noise**. As shown in Algorithm 2, compared to the Repeating Concat paradigm, this approach adds a small amount of noise to the conditional images. The intensity of the noise is not strong enough to disrupt most of the information in the conditional images, but it can enhance the generalization of the model. This paradigm is used in SVD (Blattmann et al., 2023).

**Zero-Padding Concat**. As shown in Algorithm 3, this paradigm pads the conditional image latents $s$ with zeros along the temporal dimension to match the shape of noisy latents, then concatenates them with noisy latents along the channel dimension.

**Zero-Padding Concat and Adding Noise**. As shown in Algorithm 4, compared to the Zero-padding Concat paradigm, this approach adds a small amount of noise to the conditional images. This paradigm is used in CogVideoX (Yang et al., 2024).

**Inpainting**. As shown in Algorithm 5, this paradigm treats I2V as a temporal completion task. For the conditional image $S$, the approach pads $S$ with zeros along the temporal dimension to align with the video shape. After encoding with the VAE encoder, $s$ is obtained. Additionally, a mask is generated based on $S$ to identify frames with information, which is then downsampled and rearranged to align with the frame numbers, height, and width of latents, resulting in $\mathbf{m}$. Both $\mathbf{m}$ and $s$ are concatenated with noisy latents along the channel dimension as input to the denoiser. This paradigm is used in Open-Sora Plan (Lin et al., 2024a) and Wan2.1 (Wan et al., 2025).

**Inpainting and Adding Noise**. Algorithm 6 shows that compared to the Inpainting paradigm, this paradigm first adds a small amount of noise to the conditional image, followed by temporal inpainting.

**FlashI2V (Ours)**. As shown in Algorithm 7, FlashI2V modifies only the input of the denoiser compared to other paradigms. First, the conditional image latents are encoded through a learnable projection to obtain $\phi(s)$, which acts as the shifting for the noisy latents $z_t$. Additionally, high-frequency magnitude features $s_{\text{high}}$ are extracted through the Fourier Transform. $s_{\text{high}}$ and $z_t - \phi(s)$ are concatenated together as inputs to the denoiser. According to the derivation in Sec. 3.2, we can deduce that the resulting $v$ is the velocity field conditioned on the input image.

# E   FURTHER EXPERIMENTS ON FOURIER CUTOFF FREQUENCY

In Sec. 4.4, we point out that the results obtained at lower cutoff frequencies exhibit higher fidelity, and the details in the video are more consistently preserved at inference. During training, we sample cutoff frequency percentiles from $\mathcal{U}[0.05, 0.95]$. In this section, we test the effect of using lower cutoff frequencies during training.

As shown in Fig. 8, using cutoff frequency percentiles sampled from $\mathcal{U}[0.1, 0.6]$ compared to $\mathcal{U}[0.05, 0.95]$ results in a higher probability of encountering lower cutoff frequencies, leading to a lower training loss. However, the fidelity of details in the sampling results decreases. This is because if only lower cutoff frequencies are encountered during training, the training process is dominated by Fourier guidance, and the learnable projection in latent shifting cannot be fully trained.

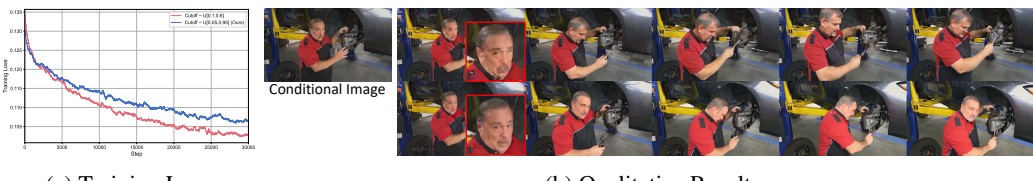

(a) Training Loss  (b) Qualitative Results

Figure 8: **The impact of different cutoff frequency percentiles during training**. (a) Using generally lower cutoff frequency percentiles results in a lower training loss. (b) Training with lower cutoff frequency percentiles leads to worse fidelity in inference. This suggests that during training, it is important to reduce the injection of high-frequency information appropriately, as an excessive input of high-frequency features can negatively affect performance.

Table 2: The source links of models and codes used in our experiments.

| Metric or Model | Source Link |
|---|---|
| FVD | https://github.com/JunyaoHu/common_metrics_on_video_quality |
| Subject Consistency | https://dl.fbaipublicfiles.com/dino/dino_vitbase16_pretrain/dino_vitbase16_pretrain.pth |
| Background Consistency | https://openaipublic.azureedge.net/clip/models/40d365715913c9da98579312b702a82c18be219cc2a73407c4526f58eba950af/ViT-B-32.pt |
| Motion Smoothness | https://huggingface.co/lalala125/AMT/resolve/main/amt-s.pth |
| Dynamic Degree | https://dl.dropboxusercontent.com/s/4j4z58wuv8o0mfz/models.zip |
| Aesthetic Quality | https://huggingface.co/sentence-transformers/clip-ViT-L-14 |
| Imaging Quality | https://github.com/chaofengc/IQA-PyTorch/releases/download/v0.1-weights/musiq_spaq_ckpt-358bb6af.pth |
| I2V Subject Consistency | https://dl.fbaipublicfiles.com/dino/dino_vitbase16_pretrain/dino_vitbase16_pretrain.pth |
| I2V Background Consistency | https://github.com/ssundaram21/dreamsim/releases/download/v0.2.0-checkpoints/dreamsim_ensemble_checkpoint.zip |
| CogVideoX1.5-5B-I2V | https://github.com/zai-org/CogVideo |
| | https://huggingface.co/zai-org/CogVideoX1.5-5B-I2V |
| Wan2.1-I2V-14B-480P | https://github.com/Wan-Video/Wan2.1 |
| | https://huggingface.co/Wan-AI/Wan2.1-I2V-14B-480P |

Table 3: **FVD values across different I2V paradigms** on training set and validation set. The quantitative results show that only FlashI2V exhibits the same FVD variation pattern on both in-domain and out-of-domain data.

(a) FVD on training set

| Method | T[0:12]↓ | T[12:24]↓ | T[24:36]↓ | T[36:48]↓ | Overall↓ |
|---|---|---|---|---|---|
| Repeating Concat | 99.27 | 133.51 | 147.12 | 145.24 | 122.00 |
| Repeating Concat and Add Noise | 89.30 | 127.12 | 142.62 | 139.85 | 119.14 |
| Zero-padding Concat | 103.37 | 144.44 | 167.21 | 166.17 | 124.50 |
| Zero-padding Concat and Add Noise | 98.86 | 137.92 | 161.54 | 157.28 | 137.03 |
| Inpainting | 104.37 | 140.39 | 155.19 | 154.20 | 131.84 |
| Inpainting and Add Noise | 116.94 | 154.23 | 170.24 | 173.45 | 148.86 |
| **FlashI2V** | **81.42** | **109.59** | **116.93** | **111.25** | **103.39** |

(b) FVD on validation set

| Method | T[0:12]↓ | T[12:24]↓ | T[24:36]↓ | T[36:48]↓ | Overall↓ |
|---|---|---|---|---|---|
| Repeating Concat | 174.86 | 187.03 | 173.08 | 172.70 | 148.92 |
| Repeating Concat and Add Noise | 163.29 | 167.10 | 150.86 | 154.84 | 143.43 |
| Zero-padding Concat | 174.74 | 182.72 | 185.05 | 183.93 | 139.28 |
| Zero-padding Concat and Add Noise | 157.28 | 165.44 | 158.16 | 159.13 | 127.68 |
| Inpainting | 197.13 | 198.75 | 198.93 | 185.20 | 151.76 |
| Inpainting and Add Noise | 251.31 | 232.13 | 210.95 | 190.42 | 156.96 |
| **FlashI2V** | **95.29** | **127.64** | **139.70** | **146.26** | **104.21** |

Table 4: **FVD values across ablation experiments** of various modules in FlashI2V on training set and validation set. The quantitative results show that latent shifting with Fourier guidance results in the best generalization and performance on out-of-domain data.

(a) FVD on training set

| Method | T[0:12]↓ | T[12:24]↓ | T[24:36]↓ | T[36:48]↓ | Overall↓ |
|---|---|---|---|---|---|
| Only Latent Shifting | 86.84 | 118.82 | 126.38 | 123.13 | 111.35 |
| Only Fourier Guidance | 159.14 | 198.89 | 210.23 | 206.68 | 158.47 |
| **FlashI2V** | **81.42** | **109.59** | **116.93** | **111.25** | **103.39** |

(b) FVD on validation set

| Method | T[0:12]↓ | T[12:24]↓ | T[24:36]↓ | T[36:48]↓ | Overall↓ |
|---|---|---|---|---|---|
| Only Latent Shifting | 107.56 | **127.31** | 136.82 | **141.06** | 113.83 |
| Only Fourier Guidance | 211.66 | 202.10 | 193.59 | 191.78 | 159.46 |
| **FlashI2V** | **95.29** | 127.64 | 139.70 | 146.26 | **104.21** |

## F  DATA COLLECTION AND PROCESSING

Our training dataset incorporates some internal data and open-source data from Panda-70M (Chen et al., 2024c) and VIDAL (Zhu et al., 2023). We also include videos from CC0-licensed websites such as Mixkit, Pexels, and Pixabay. In addition, our data collection and processing pipeline follows the Open-Sora Plan (Lin et al., 2024a).

## G  MODELS AND CODES USED IN EXPERIMENTS

As shown in Tab. 2, we provide model links to all Vbench-I2V metrics used in Sec. 4.2 and Sec. 4.3. The table also includes the link to the FVD implementation, as well as links to the CogVideoX1.5-5B-I2V and Wan2.1-I2V-14B-480P models and codes, ensuring the reproducibility of our results.

## H    MORE QUANTIFIED RESULTS

### H.1    FVD VALUES ACROSS VARIOUS I2V PARADIGMS

In Fig. 4a, to facilitate the observation of the FVD variation patterns, we present the chunk-wise FVD of different I2V paradigms in the form of a bar chart. In Tab. 3, we provide the specific values for the chunk-wise FVD and overall FVD of these paradigms. It can be observed that, compared to other I2V paradigms, FlashI2V shows consistent chunk-wise FVD variation patterns on both in-domain and out-of-domain data, and it achieves superior FVD across all chunks and overall, proving excellent generalization and performance.

### H.2    FVD VALUES ACROSS ABLATION EXPERIMENTS OF VARIOUS MODULES IN FLASHI2V

In Sec. 4.3, we present the training loss and qualitative performance for the ablation experiments of different modules in FlashI2V. Here, we provide the FVD values of the ablation experiments, as shown in Tab. 4. It can be observed that latent shifting with Fourier guidance achieved the best FVD performance, demonstrating the effectiveness of FlashI2V.

## I    MORE VISUAL RESULTS

Fig. 9 presents more inference results for FlashI2V, and the video version of the results can be found on the project page.

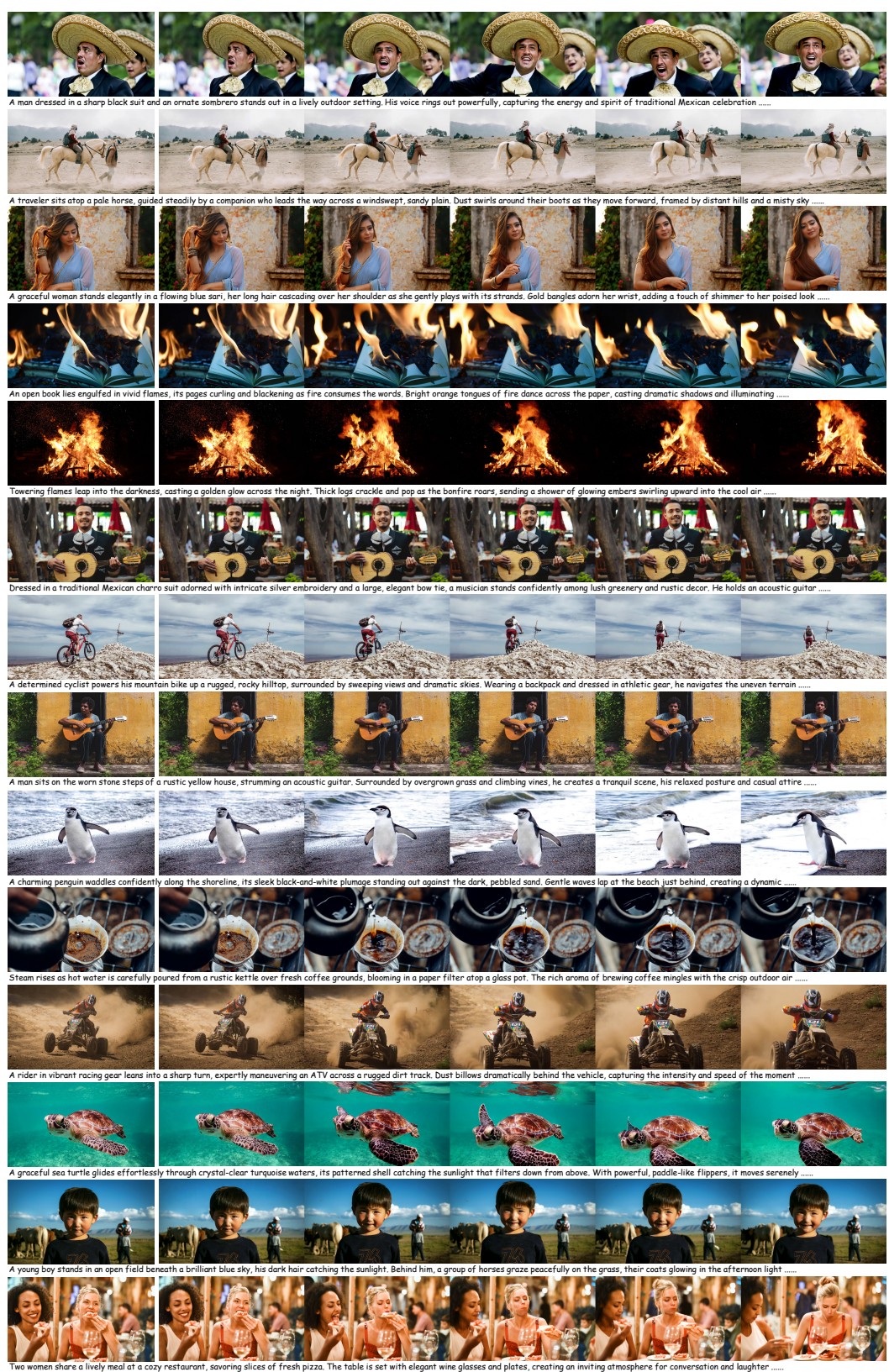

Figure 9: Visual results sampled from Vbench-I2V.

