# OpenReview forum: "FlashI2V: Fourier-Guided Latent Shifting Prevents Conditional Image Leakage in Image-to-Video Generation"
_ICLR.cc/2026/Conference — ICLR 2026 Conference Withdrawn Submission_

### Official Review · Reviewer_uPPA · 2025-10-24

**Soundness:** 2
**Presentation:** 3
**Contribution:** 2
**Rating:** 2
**Confidence:** 3

**Summary:**

The paper presents FlashI2v, an image-to-video (I2V) generation based on prompts using latent flow models. Current approaches suffer from image leakage, resulting in incoherent movements and failure to generalize to OOD contexts. The authors propose to address that issue by learning to recover a shifted version of the video from the conditional image. Also, to enhance fidelity, the authors propose to guide the process using high-frequency magnitude features. The method was evaluated on in-distribution and OOD data against other I2V methods.

**Strengths:**

* The paper deals with a timely problem in computer vision, making a coherent video generation from images. It shows some success in that, especially in the dynamic degree metric.
* Using the magnitude of the high-frequency features as a form of regularization and quality enhancement is a nice trick. As I am not aware of the literature in this area, I am not sure if it is novel or not.
* The results on OOD scenes seem strong relative to other I2V paradigms.
* The paper is written clearly and is easy to follow, although motivations for several design choices are missing as detailed next.

**Weaknesses:**

* Method:
  - Motivation to Eq. 7 is missing. While I understand why it makes sense to add the conditional image to the noise, it is less clear why to add it to $x$ as well, which should represent the full video. What should one expect when adding the same frame to all of the video frames (even if done in the latent space)?
  - If I am not mistaken, there is an error in the author's calculation of Eq. 8 and Eq. 9. Just to verify, to arrive at Eq. 8, the authors isolate $\epsilon$ in Eq. 5, and assign its value in $z_1^I$? If so, according to my calculations, additional terms are neglected. Similarly, these terms are missing in Eq. 9. Can the authors please clarify their derivation?
  - Even if the derivation in the last bullet is true, the connection to the actual proposed method in Eq. 11-14 seems contrived. If I am not mistaken, Eq. 11, 13, & 14 can all be derived by changing the boundary condition of the ODE from $x$ at $t=0$ to $x - \phi(s)$.
  - Can the authors further clarify the motivation for using $\phi(s)$ instead of $s$? What do you expect $\phi(s)$ to learn and represent? If I understand correctly, using $\phi(s)$ causes losing information about $s$ and to remedy that $s_{high}$ is used. Does that sound right?

* Experiments:
  - Overall it is hard to assess the quality of this method compared to baseline methods since the NN models are different (at least in the size, perhaps in architecture as well). For instance, it may be that increasing Flash2IV model size will result in lower dynamic degree due to higher overfitting. Conversely, it may be possible that other baselines will benefit in that metric when decreasing the number of parameters.
  - According to Table 1, the baseline SVD-XT has roughly the same number of parameters as FlashI2V and is comparable to it. Yet, this baseline is missing from all the qualitative comparisons as far as I can tell. Why is that? In which cases is FlashI2V better, and in which cases is SVD-XT? At the current state I do not see an advantage for the proposed method over this baseline.
  - It seems that the OOD experiment compares different I2V paradigms, yet not FlashI2V to different baselines. I am not sure that it shows that indeed FlashI2v is superior in OOD compared to other baselines as other methods probably have other components in them.
  - Code is missing from the submission to self-evaluate the method quality and better understand the implementation.

**Questions:**

.

---

### Official Review · Reviewer_VSGC · 2025-10-26

**Soundness:** 2
**Presentation:** 4
**Contribution:** 3
**Rating:** 6
**Confidence:** 3

**Summary:**

In Image-to-Video (I2V) generation, existing methods tend to shortcut the conditional image, a phenomenon known as conditional image leakage. This shortcutting causes the trained models to be overfit to in-domain data and struggle to generalize to out-of-domain data, leading to high Frechet Video Distance (FVD) scores for OOD generations. This paper proposes FlashI2V, which (1) subtracts the encoded conditional image from the source and target distributions during flow matching, and (2) uses high-frequency magnitude features from Fourier Transform to accelerate convergence and enable flexible control of the amount of detail in the generated video. Their results show that FlashI2V has the same FVD pattern for both IID and OOD generations, and also achieves strong performance on Vbench-I2V.

**Strengths:**

1. The problem being solved (conditional image leakage and overfitting) and the proposed solution are both well-motivated.
2. The paper is written clearly and is enjoyable to read.
3. The ablations and other control experiments were useful for understanding the underlying motivation for various components of the method.

**Weaknesses:**

1. The proposed method is not better than baseline methods, except for one metric (Table 1). It is also not significantly better than SVD (1.5B) in general.
2. The paper does not compare the training and inference speed for their method against baseline methods. Does this method take longer to train? Does this method take longer for inference?
3. The paper only evaluates on a single benchmark (Vbench-I2V). It would be helpful to evaluate on another benchmark that is complementary, i.e., evaluates something that Vbench-I2V does not evaluate.

**Questions:**

1. It would be helpful to give a short explanation for how each of the metrics (e.g., dynamic degree) in Table 1 were computed.
2. What is the FVD for the baseline methods? So we can compare against the proposed method.
3. Why do you think the proposed method improves on dynamic degree (but not other metrics on Vbench-I2V)? What is the important distinction between your method and baselines that led to this improvement?
4. The paper claims that Wan2.1 tends to produce extremely slow-motion or even static videos. Why is their dynamic degree score relatively okay, and also higher than CogVideoX1.5?
5. What do you think are the limitations of this work? What are the future steps?

---

### Official Review · Reviewer_z5F1 · 2025-10-27

**Soundness:** 3
**Presentation:** 3
**Contribution:** 3
**Rating:** 4
**Confidence:** 4

**Summary:**

This paper proposes an I2V framework that mitigates conditional image leakage, a shortcut behavior in which denoisers overly rely on the input image, leading to slow motion and color inconsistencies. The method introduces (1) Latent Shifting, which implicitly encodes the condition by subtracting image information from noisy latents in the flow matching process, and (2) Fourier Guidance, which injects high-frequency magnitude features to enhance detail and accelerate convergence. The approach achieves SOTA results on Vbench-I2V, particularly in out-of-domain generalization.

**Strengths:**

1. The paper tackles a clear technical problem in I2V generation, conditional image leakage, and gives a solid empirical analysis of its impact.
2. The combination of latent shifting and Fourier guidance is creative and seems to balance implicit conditioning with fine detail recovery quite well.
3. The experiments are thorough and well-structured, showing both in-domain and OOD performance, which adds credibility to the claims.
4. The method achieves strong results with relatively few parameters, suggesting that the design is efficient and practically relevant.

**Weaknesses:**

1. The method section is mathematically dense, and the key intuitions behind latent shifting are not clearly explained to readers unfamiliar with flow matching.
2. The role of Fourier magnitude features feels somewhat heuristic, and it is unclear why removing phase information still preserves useful detail for generation.
3. The dataset setup lacks transparency, which makes it hard to assess generalization beyond the internal distribution.
4. The ablation studies are limited; they could test more factors, such as different frequency bands or alternative conditioning forms.

**Questions:**

1. Could you clarify how latent shifting interacts with the learned flow field? Does it actually reshape the trajectory of latent transitions or just modify the starting distribution?
2. Have you explored whether Fourier guidance improves temporal coherence, not just spatial detail, given that frequency cues might encode motion energy over time?
3. Since the method implicitly removes conditioning bias, how stable is it when the input image contains elements unseen during training or with strong texture priors?
4. Would the same latent shifting idea help reduce over-conditioning in text-to-video or text-to-image diffusion models where prompt leakage is known to occur?
5. The dynamic degree metric seems to drive the main conclusion, but is it correlated with human perception of realism or motion dynamics in user studies?
6. The training dataset was collected internally, how is it reproduced?
7. How to interpretably visualize internal frequency components? Difficult to accept at the moment.

---

### Official Review · Reviewer_jUpt · 2025-10-29

**Soundness:** 2
**Presentation:** 2
**Contribution:** 2
**Rating:** 4
**Confidence:** 4

**Summary:**

This paper presents FlashI2V, a fine-tuning technique for converting text-to-video diffusion models into image-to-video (I2V) models. The paper observes that existing I2V models fine-tuned from T2V models suffer from slow motion and color inconsistency compared with their T2V counterparts. It argues that this issue arises from how existing T2V models provide the input image (i.e., the first frame) to the model — by simply concatenating it with noisy latents, which often leads to shortcut learning, where the model directly copies the input condition. To mitigate this, the paper proposes providing the input condition by subtracting it from the noisy inputs and corresponding targets. However, this approach results in slower convergence. To address this, the authors introduce Fourier Guidance, which conditions the model on high-frequency image details extracted through a Fourier transform and concatenated during training. They also remove only magnitude features (not phase) to prevent the model from generating shortcut solutions. By doing so, FlashI2V generates more dynamic videos without quality degradation compared with existing I2V models.

**Strengths:**

- The paper is generally well-written and easy to follow.
- The motivation is very clear, as empirically, people have observed I2V models tend to generate much less dynamic videos compared with generating videos through T2V models.
- The proposed method to subtract the input condition from the source/target is a simple yet interesting approach.

**Weaknesses:**

- While the paper compares the quality of generated videos with other existing I2V models, I believe the baseline should include Wan2.1 1.3B I2V models fine-tuned with common techniques (e.g., concat) using the same dataset used for fine-tuning with FlashI2V. This will remove the potential bias from pretrained models, post-training data, etc., which can validate the effectiveness of the proposed method much better.
- This paper shows a very similar motivation and goal to ALG [Choi et al., 2025]. Although ALG is a training-free method, I believe the comparison should be done in both qualitative and quantitative manners. For instance, could authors provide results that compare Wan2.1 14B+ALG with FlashI2V?
- There are several recent works (e.g., Wan2.2 5B) that try to jointly train a single model for both I2V and T2V, but it seems the proposed method cannot do.
- Figure 4 is great, but could the authors provide qualitative examples when the sample input condition and prompts are given? Because I think lower training loss might not be a good metric to support the claim made in the paper, as the model with low loss can still lead to a shortcut solution.

**Questions:**

- The proposed method only consider fine-tuning on Wan2.1; is it scalable to other T2V models, e.g., Wan2.2 or CogVideo?

---

### Note · Authors · 2025-11-14

**Comment:**

Dear reviewers, thank you for your valuable suggestions. They have helped us recognize the shortcomings of our work. We will adopt your recommendations to further improve our study. Thank you!

**Withdrawal Confirmation:**

I have read and agree with the venue's withdrawal policy on behalf of myself and my co-authors.